# Implementation of an Off-Axis Digital Optical Phase Conjugation System for Turbidity Suppression on Scattering Medium

**Kai Zhang [1], Zhiyang Wang [2],[†], Haihan Zhao [3], Chao Liu [3], Haoyun Zhang [3] and Bin Xue [3],\***

[1]  China Helicopter Research and Development Institute, Tianjin 300308, China; txzhangkai@tju.edu.cn
[2]  Science and Technology on Electro-Optical Information Security Control Laboratory, Tianjin 300308, China; wzy_sxn@tju.edu.cn
[3]  School of Marine Science and Technology, TianJin University, TianJin 300072, China; zhaohaihan@tju.edu.cn (H.Z.); achao12@tju.edu.cn (C.L.); zhanghaoyun@tju.edu.cn (H.Z.)
\*  Correspondence: xuebin@tju.edu.cn
†  Zhiyang Wang contributed equally to this work.



**Featured Application: In biomedicine, the realization of optical focusing and imaging through a scattering medium is of great significance for the internal imaging of biological tissues. Based on the study of optical focusing and imaging through a scattering medium, we can reserve related technical foundation and finally achieve internal imaging of biological tissues as well as identify diseased tissues. This work has potential application in medical diagnosis and rescue.**

**Abstract:** Due to the light scattering effect, it is difficult to directly achieve optical focusing and imaging in turbid media, such as milk and biological tissue. The turbidity suppression of a scattering medium and control of light through the scattering medium are important for imaging on biological tissue or biophotonics. Optical phase conjugation is a novel technology on turbidity suppression by directly creating phase conjugation light waves to form time-reversed light. In this work, we report a digital optical phase conjugation system based on off-axis holography. Compared with traditional digital optical phase conjugation methods, the off-axis holography acquires the conjugation phase using only one interference image, obviously saving photo acquisition time. Furthermore, we tested the optical phase conjugate reduction performance of this system and also achieved optical focusing through the diffuser. We also proved that the reversing of random scattering in turbid media is achievable by phase conjugation.

**Keywords:** off-axis holography; time reversal; digital phase conjugation; scattering medium

---

## 1. Introduction

In our daily life, there are many scattering mediums, such as milk, cloud fog, and human tissue, to name a few. The light wave surface is disturbed due to the random scattering of light in the scattering medium, so it is difficult to directly achieve optical focusing and imaging for its internal targets, which is the main restricting factor for imaging depth and resolution in medical diagnosis and treatment [1–5]. In order to control the scattered wavefront and suppress the light scattering, researchers proposed wavefront shaping [6–9], transmission matrix [10–13], phase conjugate [14–26], and other techniques. Among them, the optical phase conjugate technique uses the phase conjugate modulation of the optical wave surface to realize the time reversal of the light wave, so that the wavefront distortion is compensated in a targeted manner, and the scattering effect is suppressed to achieve focusing as

well as imaging. The response speed of the optical phase conjugate technique is fast, and its real-time performance is strong, and these features make it the current research hotspot in related fields [14–26].

Generally, the optical phase conjugation can be divided into analog modulation [14–16] and digital modulation [17–26]. Analog modulation uses crystals as an optical modulation as well as recovery medium. Digital modulation realizes optical phase recording and recovery by a camera and a spatial light modulator (SLM). Compared with analog modulation, digital modulation has stronger controllability, higher sensitivity, shorter exposure time, and it also has no wavelength and light intensity limitation, so that it has wider application prospects. In 2010, Cui et al. first established the digital optical phase conjugation (DOPC), which used a four-step phase-shifting method to measure the speckle phase and realized the phase conjugate focusing through the scattering medium [17]. Since then, DOPC has been deeply researched for focusing imaging of the scattering medium. In 2012, Wang et al. reduced the alignment complexity of a COMS (complementary metal oxide semiconductor) camera and SLM by improving the system design of DOPC, and fluorescence imaging inside the scattering medium was also realized by using ultrasonic modulation [18]. In the same year, Si et al. also realized fluorescence imaging of the isotropic medium in the three-dimensional restricted area of light [19]. Vellekoop et al. presented a reference-free digital optical phase conjugation (DOPC) method that is sensitive enough to detect very weak fluorescent signals and works for light that is both spatially and temporally incoherent [20]. In 2013, Hillman et al. presented a simple and robust digital optical phase conjugation implementation for suppressing multiple scattering. They experimentally demonstrated wide-field imaging through a highly scattered sample using DOPC employing a ring interferometer configuration [21]. In 2014, Suzuki et al. realized the ultrasonic-encoded digital optical phase conjugate in the reflection mode for the first time, further promoting the practical application of the DOPC system [22]. In 2015, Lee et al. demonstrated the realization of a one-wave optical phase conjugation mirror using a spatial light modulator. Their method is simple, alignment free, and fast while allowing high-power throughput in the time reversed wave [23]. In the same year, Ruan et al. presented a new technique, known as time-reversed ultrasound microbubble encoded (TRUME) optical focusing, which focuses light with improved efficiency and sub-ultrasound wavelength resolution [24]. In 2016, Ryu et al. further realized the variable focus and three-dimensional imaging of transmitting scattering medium, which improved the controllability of phase conjugate focusing imaging [25]. In 2017, Liu et al. used a ferroelectric liquid crystal-based spatial light modulator to develop a simple but fast DOPC system that focuses light not only through, but also inside, scattering media. They employed a ferroelectric liquid crystal-based SLM to achieve binary-phase modulation for high speed and high focusing quality [26]. At present, some scholars in China are interested in researching the relevant performance of DOPC from theoretical and practical aspects [27–32].

However, most of the above-mentioned DOPC systems use four-step phase-shifting holography to extract the speckle phase. This method requires four interferograms to be acquired continuously in the experiment. The phase shift of the corresponding reference light is 0, $\pi/2$, $\pi$, and $3\pi/2$, and the acquisition time of the images greatly limits the time reversal efficiency of the digital optical phase conjugate system.

In this paper, an optical phase conjugate system based on off-axis digital holography is designed. The phase distribution of the speckle is calculated by collecting only one interference image at a time. Compared with the four-step phase shift method, the phase acquisition time is greatly shortened, and the optics phase conjugate efficiency is also improved. At the same time, we tested the reduction performance of the phase conjugate system, and optical focusing through the diffuser is achievable.

## 2. Experimental Principle

The optical phase conjugate is used to focus through a scattering medium. The basic principle is to realize the time reversal of the light by calculating the phase of the speckle light and taking the conjugate thereof, that is, returning along the original path. Usually, an ideal plane light wave is expressed as $Ae^{-j\phi}$ (ignoring the optical frequency term); after passing through the scattering medium,

the amplitude and phase of the wavefront are modulated, and the speckle field formed on the latter plane is expressed as $A\rho(x,y)e^{-j(\phi+\varphi(x,y))}$. If we get $\rho(x,y)$ and $\varphi(x,y)$, we achieve phase and amplitude modulation of the light wave, so that the emitted light can be expressed as $B\rho(x,y)e^{-j(\phi+\varphi(x,y))}$, which means time reversal of the light wave can be achieved to suppress the scattering effect of the light. Theoretical analysis and practical research show that, compared with the combined modulation of phase and amplitude, only phase modulation also effectively suppresses the scattering effect [33]. In this paper, pure phase SLM is used, and we only study the phase modulation of the optical wave surface.

The experimental setup designed in this paper is based on the off-axis holographic DOPC system (as shown in Figure 1), and the phase of the light field is extracted by the interference of the object light with angular difference and the reference light on the surface of the SLM [34].

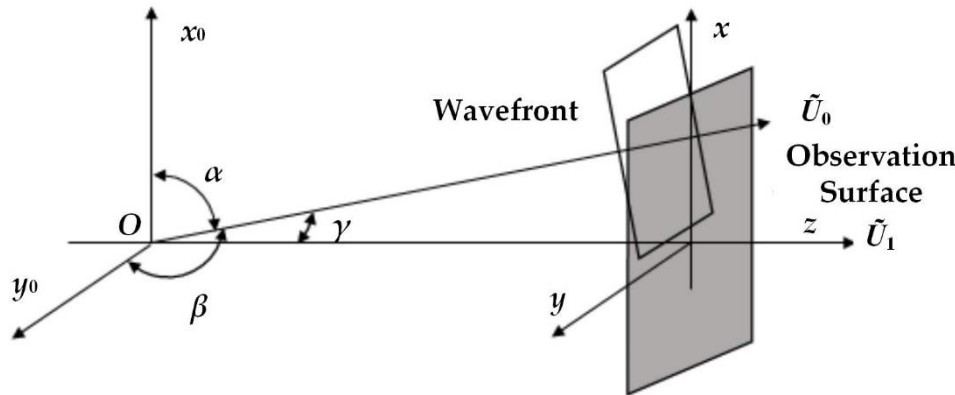

**Figure 1.** Off-axis holographic interference.

As shown in Figure 1, the spatial coordinate system is established, which uses the intersection of the two planes light waves, $\tilde{U}_0$ and $\tilde{U}_1$, as the origin. It is assumed that the plane light wave $\tilde{U}_0$ is injected at a specific angle with $x$ and $y$ axes, and $\tilde{U}_1$ is perpendicular to the observation surface, the distance between the observation surface. If $\tilde{U}_1$ is $z$, then the ideal off-axis holographic interference intensity distribution $U(x,y,z)$ is obtained on the observation surface.

$$\widetilde{U}_0(x,y,z) = U_0(x,y,z)\exp\left(j\frac{2\pi}{\lambda}z\sqrt{1-\cos^2\alpha-\cos^2\beta}\right)\exp\left[j\frac{2\pi}{\lambda}(x\cos\alpha+y\cos\beta)\right] \tag{1}$$

$$\widetilde{U}_1(x,y,z) = U_1(x,y,z)\exp\left(j\frac{2\pi}{\lambda}z\right) \tag{2}$$

$$\begin{aligned} U(x,y,z) &= \left(\widetilde{U}_0+\widetilde{U}_1\right)\times\left(\widetilde{U}_0+\widetilde{U}_1\right)^* \\ &= U_0^2 + U_1^2 + 2U_0U_1\cos\left[\frac{2\pi}{\lambda}\left(x\cos\alpha+y\cos\beta+z\sqrt{1-\cos^2\alpha-\cos^2\beta}-z\right)\right] \end{aligned} \tag{3}$$

We assume that $\beta = \pi/2$, when $(\pi/2\lambda)x\cos\alpha + z(\sin 2\alpha)^{1/2} - z = 2k\pi$, ($k = Z$, $Z$ represents a positive integer), we get the maximum value of $U$; when $(\pi/2\lambda)x\cos\alpha + z(\sin 2\alpha)^{1/2} - z = (2k+1)\pi$, ($k = Z$, $Z$ represents a positive integer), we get the minimum value of $U$. Thus, the spacing between adjacent bright (dark) stripes is expressed as:

$$\Delta = \frac{\lambda}{\cos\alpha} \tag{4}$$

In the actual experiment, the distance $z$ is fixed and regarded as a constant omitted in the formula. Considering the two-dimensional plane only, we assume that the reference light is a uniform plane light whose initial phase is $\psi$, and it is perpendicular to SLM plane; thus, the light field distribution on the SLM is expressed as:

$$\widetilde{R}(x,y) = A(x,y)e^{-j\psi} \tag{5}$$

By making the object light on XY plane inject to the SLM plane at an angle $\alpha$ with the *X*-axis direction, the object light wave surface is expressed as:

$$\widetilde{O}(x,y) = B(x,y)e^{-j2\pi\xi x}e^{-j\phi(x,y)} \tag{6}$$

where $\xi = \cos \alpha/\lambda$ and $\lambda$ is the wavelength of the light.

We used a complementary metal oxide semiconductor (CMOS) camera to collect the light intensity information of the SLM surface. Considering the two-dimensional sampling theorem, the parameter $\alpha$ in the above formula needs to satisfy the below condition:

$$\alpha \leq \alpha_{\max} = \arccos(\lambda/2\Delta x) \tag{7}$$

where $\Delta x$ is the pixel size of the camera. Its light intensity distribution is expressed as:

$$
\begin{aligned}
I(x,y) &= \left|\widetilde{R}(x,y) + \widetilde{O}(x,y)\right|^2 \\
&= \left|\widetilde{R}\right|^2 + \left|\widetilde{O}\right|^2 + \widetilde{R}^*\widetilde{O} + \widetilde{R}\widetilde{O}^* \\
&= \widetilde{U}_1(x,y) + \widetilde{U}_2(x,y) + \widetilde{U}_3(x,y) + \widetilde{U}_4(x,y)
\end{aligned} \tag{8}
$$

After two-dimensional Fourier analysis of the light intensity distribution image, it is easy to obtain:

$$
\begin{aligned}
F\{\widetilde{U}_1(x,y)\} &= \delta(u,v) \\
F\{\widetilde{U}_2(x,y)\} &= \widetilde{G}_0(u,v)\widetilde{G}_0(u,v) \\
F\{\widetilde{U}_3(x,y)\} &= \widetilde{G}_0(u-\alpha,v) \\
F\{\widetilde{U}_4(x,y)\} &= \widetilde{G}_0(-u-\alpha,-v)
\end{aligned} \tag{9}
$$

where $\widetilde{G}_0(u,v) = F\{\widetilde{O}(x,y)\}$ is the spectrum of the speckle light field, $F$ stands for two-dimensional Fourier transform, $\star$ represents a correlation operation, and $\delta(u,v)$ is an impulse function.

$$F\{I(x,y)\} = \delta(u,v) + \widetilde{G}_0(u,v)\widetilde{G}_0(u,v) + \widetilde{G}_0(u-\alpha,v) + \widetilde{G}_0(-u-\alpha,-v) \tag{10}$$

It can be seen from the two-dimensional Fourier analysis that $\tilde{U}_1$, $\tilde{U}_2$, $\tilde{U}_3$, and $\tilde{U}_4$ appear on different positions of the frequency spectrum, respectively. The appropriate two-dimensional band pass filter is used to extract the spectrum which contains the speckle light field $\widetilde{G}_0(-u-\alpha,-v)$ separately, and then the amplitude and phase of the speckle light field are obtained through performing two-dimensional Fourier inverse transformation. That is, we use $F^{-1}\{\widetilde{G}_0(-u-\alpha,-v)\} = \tilde{U}_4(x,y)$ to obtain $\widetilde{R}\widetilde{O}^*$, which makes the phase distribution apply to the SLM, and also makes the reference light inject to the SLM at a conjugate angle to obtain a conjugate phase light field of the speckle light field. When the reference light is perpendicular to the observation surface, the conjugate light is also perpendicular to the observation surface, so it is only necessary to keep the reference light to inject perpendicularly during the phase conjugate reduction process.

## 3. Materials and Methods

We designed and built the optical phase conjugate system based on off-axis digital holography. The phase conjugate performance of the system was tested. The experimental schematic diagram is shown in Figure 2, and the experimental process is mainly divided into two parts.

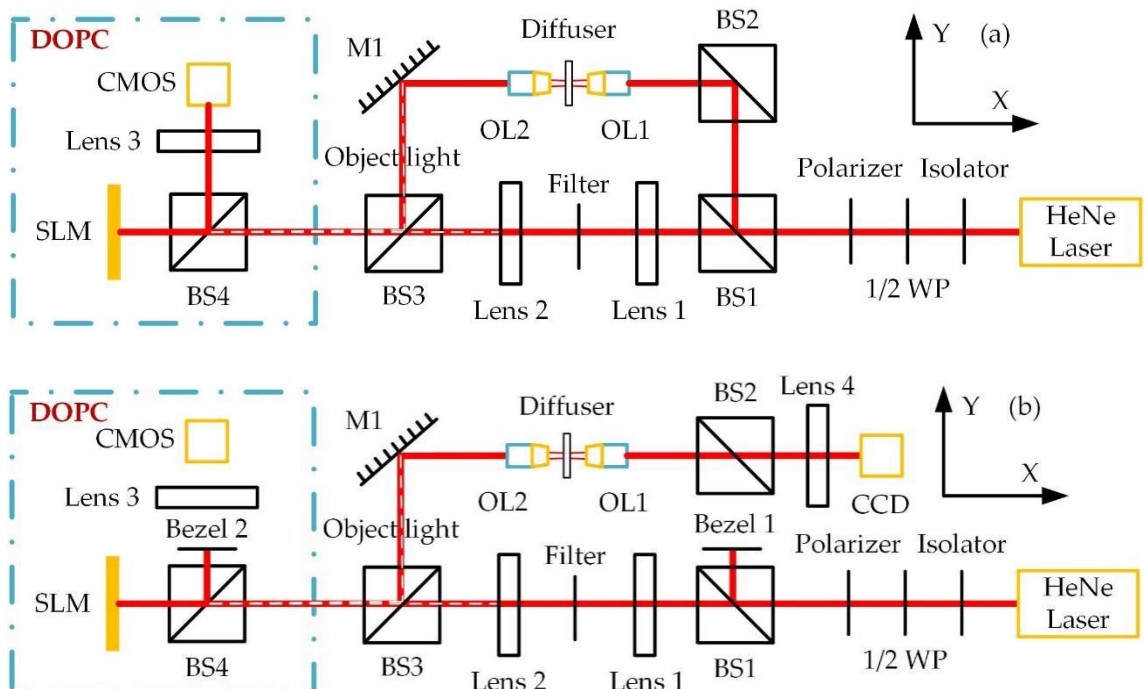

**Figure 2.** Experimental setup: (**a**) phase extraction; (**b**) time reversal. WP: waveplate; BS: beam splitter; M1: mirror; SLM: spatial light modulator; DOPC: digital optical phase conjugation; OL: objective lens; CMOS: complementary metal oxide semiconductor camera.

### 3.1. Phase Extraction

As shown in Figure 2a, we used a HeNe laser (Thorlabs, Newton, NJ, USA, HRS015B) working in a stable mode. After the laser beam emitted by a HeNe laser passed through the optical isolator, 1/2 wave plate, and polarizer, we adjusted the polarization to a horizontal direction to ensure the maximum modulation efficiency of the SLM (Holoeye, Berlin, Germany, PLUTO-VIS-016, resolution 1920 pixels × 1080 pixels, pixel size 8 μm × 8 μm). Then, beam splitter 1 was used to divide the incidence beam into two perpendicular laser beams: one beam passed through beam splitter 2, objective lens 1, diffuser (Thorlabs, DG100X100-120, Ground Glass) and objective lens 2, and at last was reflected by a mirror and injected into beam splitter 3; as a reference light, the other beam passed through lens 1 and filter and lens 2, the beam diameter became 18 mm, and at last the object light and reference light were combined at beam splitter 3 to enter the DOPC system together. In our experiment, the reference light was injected perpendicularly into the SLM, and the object light was injected at a specific angle with the reference light. We used a CMOS camera in the DOPC system to collect the interference patterns on the SLM, and then we performed spectrum analysis of the patterns collected before. Finally, we obtained the conjugate phase of the speckle light field by frequency domain filtering and inverse Fourier transform calculation.

The key to digital optical phase conjugation based on off-axis holography is to separate the spectral information which contains the phase of the object light field. According to Equation (4), the larger the angular deviation of the two beams (the smaller $\alpha$), the smaller the spacing of adjacent bright (dark) stripes and the denser the interference fringes. In the case where the resolution of the camera is not exceeded, that is, Equation (7) is satisfied, the angular difference between the reference light and the object light increases as much as possible to ensure that the optical spectrum of each stage is separated as much as possible during the Fourier transform. Furthermore, according to Equation (9), as long as the relative incident angle of the reference light and the object light is kept constant, the distance between the object light field and the central light field in the frequency domain remains unchanged. Therefore, by fixing the relative angle of the reference light and the object light and setting a suitable

spatial filter around the object light field in the frequency domain, the phase of the object light field is better restored.

### 3.2. Phase Modulation and Time Reversal

The time reversal process is shown in Figure 2b. After placing the bezel shown in Figure 2b, we adjust the intensity of the reference light and finally load the speckle conjugate phase on the SLM. The reference light was phase-modulated by the SLM to form a light wave conjugated with the original speckle phase, that is, conjugate compensation for the scattered light to form a time-reversed light wave, which reversely passed through the diffuser and then was detected by a CCD (charge coupled device) camera.

The DOPC system shown in Figure 2 consists of a SLM (Holoeye, PLUTO-VIS-016, resolution 1920 pixels × 1080 pixels, pixel size 8 μm × 8 μm), a beam splitter, a CMOS camera (PCO.EDGE 3.1, resolution 2048 pixels × 1536 pixels, pixel size 6.5 μm × 6.5 μm), and a lens (Lens 3, f = 100 mm). The SLM is mounted on a six-degrees-of-freedom combination mount. Since the pixel size of the SLM and CMOS camera do not match, it is necessary to adjust the Lens 3 so that the SLM and CMOS camera sensor planes are mutually imaged [17], to ensure the SLM and COMS camera pixels are aligned one by one; thus the camera can accurately extract the interference patterns on the SLM. According to the theoretical calculations, the object image relationship of the convex lens satisfies $u_o + v_i = f \cdot (M + 1)^2/M$ ($u_o$: object distance, $v_i$: image distance, f: focal length, M: magnification). We determined the target magnification is 0.8125 (M = 6.5/8) from the pixel size of the SLM and CMOS camera. In the experiment, the Lens 3 focal length is 100 mm, so the spatial distance between the CMOS camera and SLM needs to be adjusted to approximately 404 mm.

The pixel matching process of the SLM and CMOS camera is divided into two steps: coarse adjustment and fine adjustment. We removed the Lens 3 in front of the CMOS camera during coarse adjustment and adjusted the position of the SLM, CMOS camera, and beam splitter 4 to place the light not only at the center of the SLM and CMOS camera, but also incident vertically. We loaded a phase pattern with a specific pattern on the SLM and adjusted its imaging focal length. When a clear pattern is presented on a CMOS camera, this focal length is the spatial straight-line distance from the SLM to CMOS camera. We adjusted the position of the SLM and CMOS camera to ensure that the two devices were center-aligned horizontally. Before fine adjustment, we placed Lens 3 in front of the CMOS camera and adjusted its spatial position to ensure that its focus was at the center of the CMOS camera. We then loaded the grid image on the SLM and adjusted the distance between Lens 3 and the CMOS camera to make a clear image on the latter. We observed the magnification corresponding to the grid, if M > 0.8125, it is necessary to increase the spatial distance between the CMOS camera and SLM, as well as adjust the position of Lens 3 again to ensure the CMOS camera presents a clear image. We repeated the operation several times to guarantee magnification in the condition that the imaging is clear. After the above-two steps, the pixel matching process for the SLM and CMOS camera was completed.

In the experiment, the phase conjugate reduction performance of the off-axis holographic DOPC system without the scattering sheet was tested, and then the scattering sheet was placed to verify its ability to focus through the scattering medium.

## 4. Experimental Results and Discussion

### 4.1. Time Reversal Experimental Results

Figure 3a is an off-axis holographic interference pattern acquired by a CMOS camera when the diffuser is not used during the phase extraction process. The spacing of the light (dark) fringes is about 28 pixels, which satisfies the two-dimensional sampling theorem of the image. After two-dimensional Fourier transform, the spectrogram is obtained (see Figure 3b), and it is clearly visible that the spectrogram is divided into three regions. According to the off-axis digital holography theory, the three

regions in the spectrogram correspond to $\tilde{U}_3$, $\tilde{U}_1 + \tilde{U}_2$, $\tilde{U}_4$ in Equation (9) from left to right. As shown in Figure 3c, a suitable frequency domain filter is constructed to only make the spectrograms with $\widetilde{RO}^*$ left. The inverse Fourier transform of the spectrograms left obtains $\widetilde{RO}^*$, and its phase distribution is shown in Figure 3d. During the time reversal, the conjugate phase pattern was loaded on the SLM, and the conjugate reference light was injected to obtain a light wave $\widetilde{O}^*$ which was conjugated with the original optical phase; thus, the time reversal of the object light is realized.

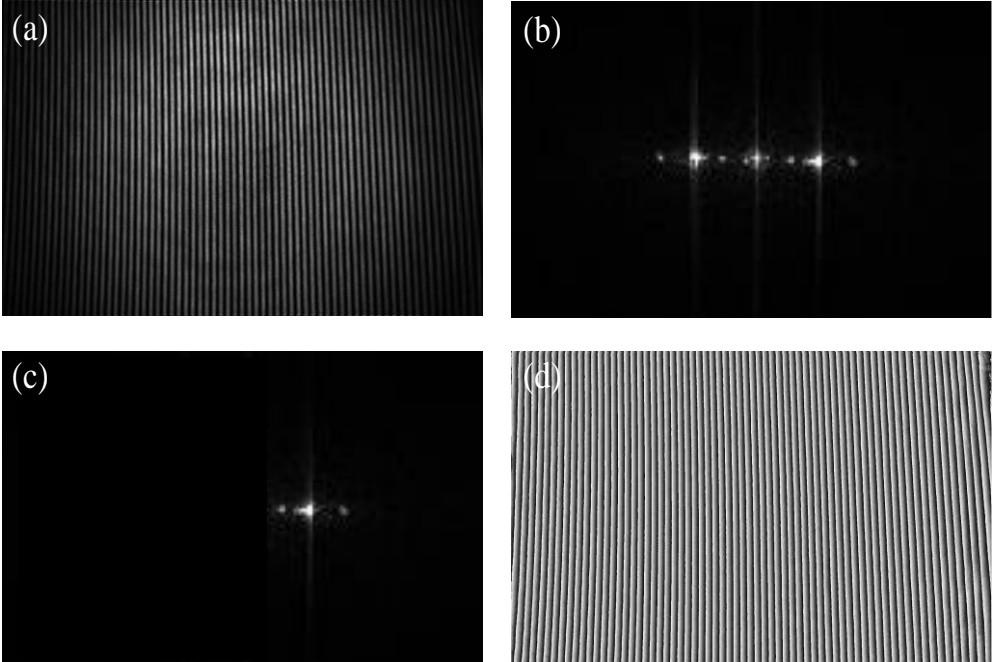

**Figure 3.** (**a**) Interference pattern; (**b**) spectrum of the interference pattern; (**c**) spectrum after filtering; (**d**) phase conjugation pattern.

After completing the basic construction of the off-axis digital holographic optical phase conjugate system, the performance of the system to achieve light wave time reversal was tested. After controlling the two-dimensional precision stage to move OL2 (objective lens 2) along the *X*-axis and *Y*-axis (see Figure 2) respectively, the influence of the propagation path variation of the object light to the phase conjugate reduction was tested. Moving OL2 along the *X*-axis with a step size of 50 μm changes the collimation ability of the object light. When the reference light is directly reflected from the SLM to be modulated, the spot size on the CCD changes. If the conjugate phase of the object light is loaded on the SLM, the light will return along the original path and form a time-reversed light wave of the object light, thus the size of the spot detected on CCD remains the same. Figure 4b is a curve which shows the relationship between the lights spot size recorded by the CCD and the displacement along the *X*-axis. It can be seen from the figure that when the displacement along the *X*-axis is in the range of (−350 μm, 350 μm), the conjugate light achieves a better phase conjugate reduction effect than the direct reflected light. In addition, moving OL2 along the *Y*-axis with a step size of 10 μm changes the spot position of the unmodulated light on the CCD. Figure 5b shows the spots position change of the unmodulated reflected light as well as the time-reversed conjugate light on the CCD. According to the phase conjugation theory, the time-reversed modulated light can return along the original path, so ideally, the position of the spot detected on the CCD does not change with the change of the position of OL2. It can be seen from Figure 5b that the off-axis digital holographic phase conjugate system has a good phase conjugate reduction effect when the displacement along the *Y*-axis is in the range of (−80 μm, 80 μm).

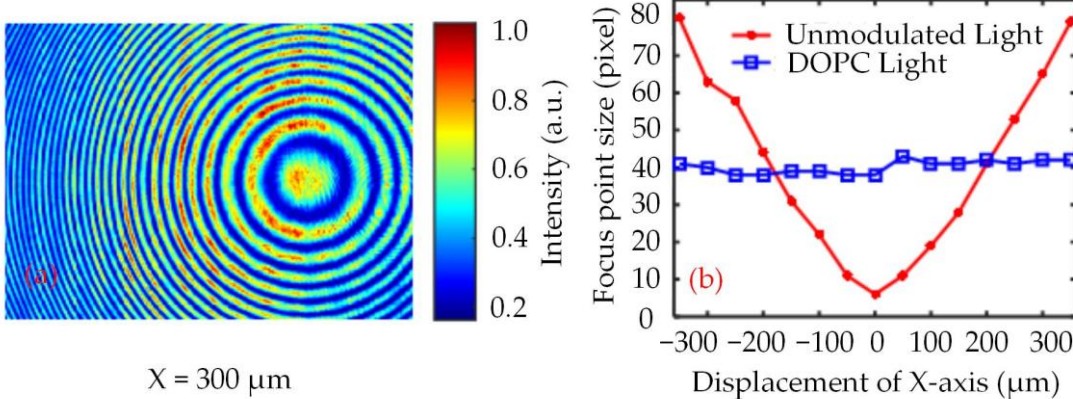

**Figure 4.** (**a**) Interference pattern when $X = 300$ μm; (**b**) the reflected focus and reconstructed focus size variation when OL2 was shifted by the $X$-axis.

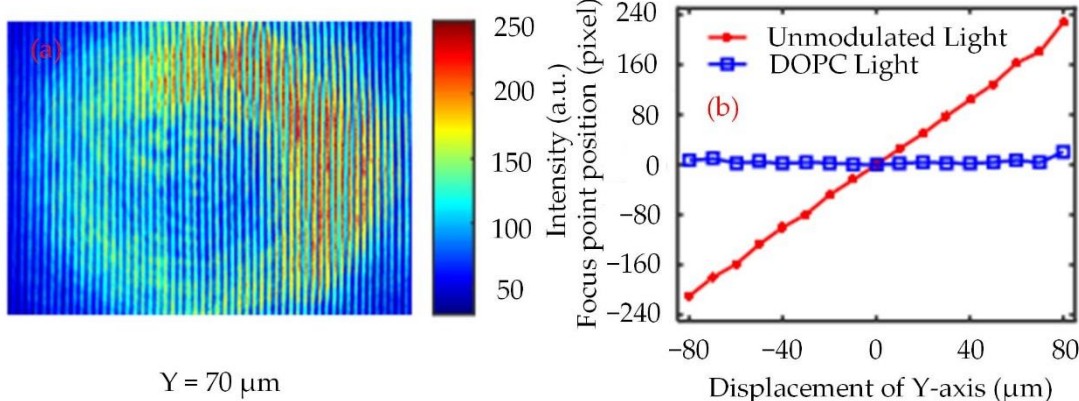

**Figure 5.** (**a**) Interference pattern when $Y = 0.8$ mm; (**b**) the reflected focus and reconstructed focus position variation when OL2 was shifted by the $Y$-axis.

During the experiment, we adjusted the relative position of Lens 3 and the CMOS camera, and also changed the imaging resolution of the CMOS camera to the SLM to observe the influence of those actions above the phase conjugate reduction quality. Figure 6 shows a curve of the intensity of the spot collected by CCD changes as a function of the displacement of Lens 3. As can be seen from the figure, when the displacement error of Lens 3 is within the range of ($-1$ mm, 1 mm), the intensity does not change much. However, as the position of Lens 3 moves farther away from the center point, the CMOS imaging becomes less clear, and the spot intensity at the focal point decreases at a certain extent. The figure shows that although the influence of the image definition on phase conjugate reduction is limited within a certain range, if we want to achieve high-quality phase conjugate reduction, the requirement of clear imaging of the DOPC system is indispensable.

In order to further analyze the influence of phase noise to the phase conjugate reduction of light waves, random phase noise was artificially added to the calculated conjugate phase diagram. We controlled the intensity range of the random phase noise to test the resistance to phase noise of the DOPC system. Figure 7 shows the relationship between the spot intensity and phase noise. It can be seen from Figure 7 that the DOPC system still has a good conjugate phase reduction performance when the phase noise is in the range of $(0, \pi/2)$. When the phase noise exceeds $\pi/2$, the phase conjugate reduction performance of the DOPC system decreases rapidly, so it is difficult to effectively realize the time reversal of the light wave.

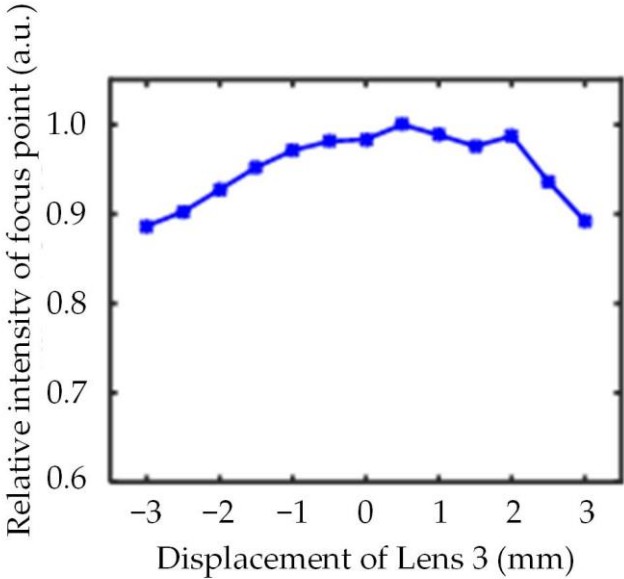

**Figure 6.** Reconstructed focus intensity variation as change of clarity.

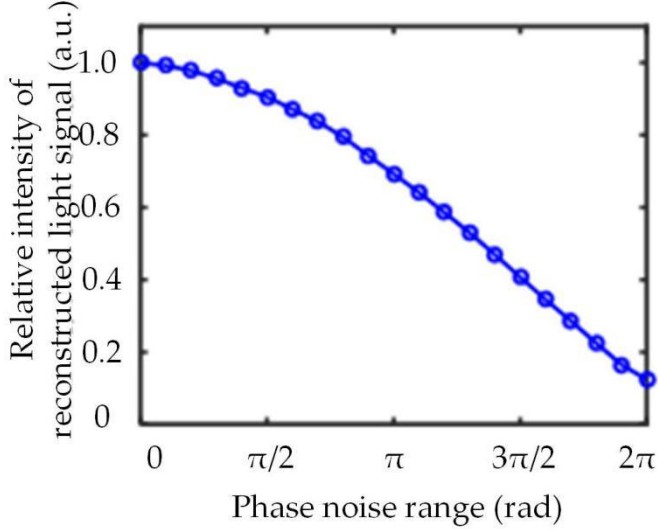

**Figure 7.** Experimental measurements of the reconstructed DOPC signal dependence on the amount of phase error.

### 4.2. Experimental Results of the Ability to Recover the Scattered Light Field

In order to test the ability of the phase conjugate system for recovering the scattered light field, the diffuser was placed at the focus of both OL1 and OL2. Figure 8a shows the off-axis speckle interference pattern acquired by the CMOS camera. It can be seen from the figure that the plane light became a random scattered light wavefront after passing through the scattering medium. Figure 8b is a calculated off-axis speckle conjugate phase diagram. Figure 8c shows the imaging acquired by the CCD when the phase of the SLM was set to 0, that is, the DOPC system was not used. At this time, the scattering medium is an unmodulated plane light wave, so the outgoing light was a random speckle pattern. Figure 8d shows the pattern acquired by the CCD when the conjugate phase was loaded on the SLM, that is, the DOPC system was used. At this time, the scattering medium was preceded by a conjugate wavefront, which formed a time-reversed outgoing light and also formed a focal point on the CCD. The size of the focal point mentioned above was about 220 μm, compared with the nearly

52 μm point size in the case where the diffuser was not used, we conclude that the phase conjugate modulated light achieves a good focusing effect after passing through the diffuser. At the same time, by analyzing the experimental data, we determined that the average intensity of the background noise and focal point in Figure 8d is about 150 and 54,750, respectively. Further, the peak-to-background intensity ratio between the focal point and the background in Figure 8d is about 365.

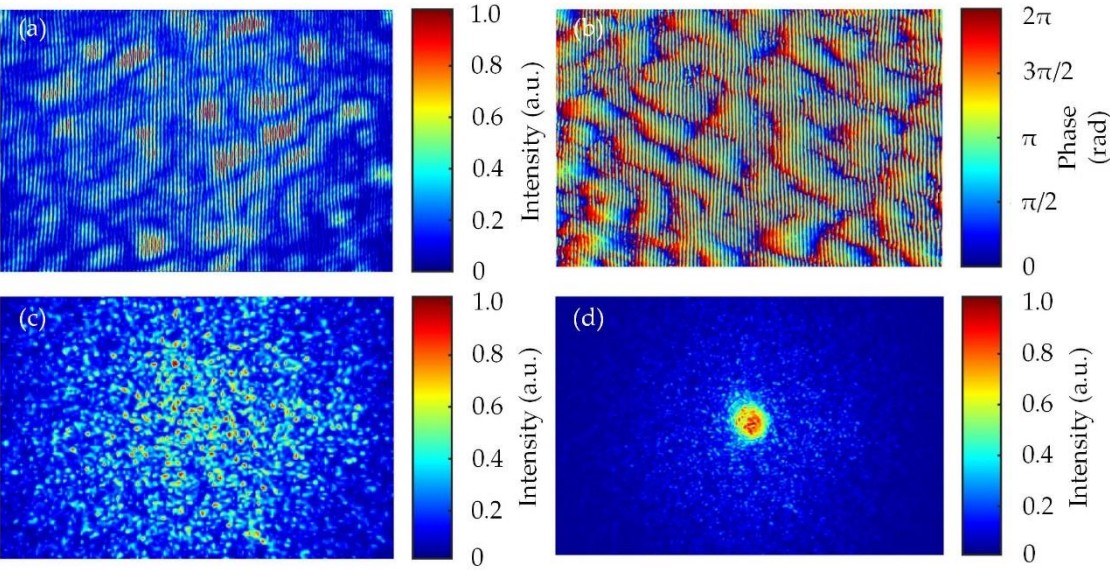

**Figure 8.** (**a**) Speckle interferogram; (**b**) phase conjugation pattern; (**c**) charged couple device's (CCD) signal with the phase of the SLM set to 0; (**d**) DOPC reconstrued signal with loading conjugation phase.

## 5. Conclusions

A phase conjugate system based on off-axis digital holography was designed. Based on this system, we achieved optical focusing of the light which passes through a scattering medium. The experimental results show that the phase conjugate system has good phase reduction performance, and even when the object light changes, the phase conjugate reduction also performs well. In addition, the influence of image definition as well as phase noise to time-reversed light was tested. The experiments show that the system has a certain fault tolerance. The optical phase conjugate focused on transmitting the scattering medium was successfully realized even though the diffuser was added to the system. The off-axis digital holographic phase conjugate system obtains the phase of the object light field only by acquiring one interferogram, which greatly improves the efficiency of the phase conjugate system compared with the four-step phase-shifting method. This work also promotes the practical uses of the digital optical phase conjugate system. However, since the off-axis digital holography depends on the angular difference between the object light and reference light, the range of the phase recovery point still has some limitations because of the limitation of the camera resolution. In addition, there will inevitably be some distortions in the imaging between the SLM and CMOS camera. Studying and correcting the distortion helps to further improve the reduction capability of the digital phase conjugate system.

**Author Contributions:** B.X. came up with this idea. K.Z. and Z.W. designed the experiment. K.Z., Z.W., H.Z. (Haihan Zhao), C.L., and H.Z. (Haoyun Zhang) performed the experiment. K.Z. and Z.W. analyzed the data. K.Z. and Z.W. wrote the original manuscript. B.X., H.Z. (Haihan Zhao), and C.L. did the review and editing. All authors have read and agreed to the published version of the manuscript.

**Funding:** This research was funded by the Projects from Tianjin University of Technology and Education, grant number KJ15-24; the Natural Science Foundation of Tianjin, grant number 18JCYBJC17100; the Natural Science Foundation of Tianjin, grant number ZZSK021917.

**Conflicts of Interest:** The authors declare no conflicts of interest.

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
