# Peer review of "Implementation of an Off-Axis Digital Optical Phase Conjugation System for Turbidity Suppression on Scattering Medium"

_applsci, doi:10.3390/app10030875_

Round 1
Reviewer 1 Report
I read the manuscript applsci-700689 titled “Implementation of an off-axis digital optical phase conkucation system for turbidity suppression on scattering medium” by K. Zhang et alii that was submitted to Applied Sciences for publication. The manuscript deals with a new experimental configuration to compensate spatial dispersion in ultra-dense media and to allow imaging through them.
The experimental configuration is proper and the theoretical model to describe the experimental set-up is well fitted. The manuscript is free from fundamental errors and can be published after a review of the English language by a native English speaker.
Reviewer 2 Report
Zhang et al. present a method for digital optical phase conjugation (DOPC) using an off-axis interferometric field measurement and wavefront shaping using one spatial light modulator (SLM). The concept of DOPC is not new, but as the authors tried to address, the implementation of DOPC is challenging. This is mainly because the measurement and modulation of optical fields have been done using separate optical paths in previous approaches or using requiring a reference beam for holography. In that view, the authors proposed a clear idea of using an SLM for both off-axis holography and wavefront shaping. The method is sound and the results are free from obvious mistakes. Thus, in principle, I support the publication of the manuscript, provided that the following issues are addressed.
Previously, the simplified setups for DOPM had been published, but not cited and discussed in the current manuscript. For example, a Sagnac-based optical setup was proposed [Hillman, Timothy R., et al. "Digital optical phase conjugation for delivering two-dimensional images through turbid media." Scientific Reports 3 (2013): 1909]. More recently, a simpler setup was also published [Lee, KyeoReh, et al. "One-wave optical phase conjugation mirror by actively coupling arbitrary light fields into a single-mode reflector." Physical review letters 115.15 (2015): 153902.]
Also, there are other highly relevant papers that were not mentioned in the manuscript, including Ruan, Haowen, Mooseok Jang, and Changhuei Yang; Liu, Yan, et al. "Focusing light inside dynamic scattering media with millisecond digital optical phase conjugation." Optica 4.2 (2017): 280-288. "Optical focusing inside scattering media with time-reversed ultrasound microbubble encoded light." Nature communications 6 (2015): 8968.
The quality of the DOPC results presented in the current manuscript does not meet the quality of the state-of-the-art demonstrations in the previous DOPC papers (see the results in the mentioned papers above). Even though enhancing the quality of the DOPC results is not the main point of the manuscript, I would like to suggest the authors quantitatively analyzing and discussing the results. For example, the authors may want to quantify the size of the beam in Fig. 8(d) and compare with the expected (theoretical) values or the case in the absence of a scattering layer. Also, the intensity ratio of the peak compared to the background in Fig. 8(d) can be investigated with the controlled modes (number of active pixels) in the used SLM. Alternatively, the circular averaged intensity profile of Fig. 8(d) can be compared to that of Fig. 8(c) – please refer to the results in Ref. 3.
Please add the scale bars for all the relevant figures.
Please provide detailed information about the used diffuser (e.g., scattering parameters or diffusion angles, model number, manufacturer).
There are many minor English usage/syntax problems -- too many to list here. It's generally clear what the authors are trying to get across, but the errors are distracting. To make sure that the paper and its results are accessible to the majority of readers, the paper should be revised to conform to standard English grammar.
Appendix A is not necessary.
